# Genome-Wide Identification and Expression Analysis of GA2ox, GA3ox, and GA20ox Are Related to Gibberellin Oxidase Genes in Grape (*Vitis*
*vinifera* L.)

**DOI:** 10.3390/genes10090680

**Published:** 2019-09-05

**Authors:** Honghong He, Guoping Liang, Shixiong Lu, Pingping Wang, Tao Liu, Zonghuan Ma, Cunwu Zuo, Xiaomei Sun, Baihong Chen, Juan Mao

**Affiliations:** 1College of Horticulture, Gansu Agricultural University, Lanzhou 730070, China; 2College of Resource and Environmental Sciences, Gansu Agricultural University, Lanzhou 730070, China

**Keywords:** Grape, Gibberellin oxidase, gene family identification, codon bias, Gene duplication, Abiotic stress, qRT-PCR

## Abstract

Gibberellin (GAs) plays the important role in the regulation of grape developmental and growth processes. The bioinformatics analysis confirmed the differential expression of GA2, GA3, and GA20 gibberellin oxidase genes (*VvGA2oxs*, *VvGA3oxs,* and *VvGA20oxs*) in the grape genome, and laid a theoretical basis for exploring its role in grape. Based on the Arabidopsis *GA2oxs*, *GA3oxs*, and *GA20oxs* genes already reported, the *VvGA2oxs*, *VvGA3oxs*, and *VvGA20oxs* genes in the grape genome were identified using the BLAST software in the grape genome database. Bioinformatics analysis was performed using software such as DNAMAN v.5.0, Clustalx, MapGene2Chrom, MEME, GSDS v.2.0, ExPASy, DNAsp v.5.0, and MEGA v.7.0. Chip expression profiles were generated using grape Affymetrix GeneChip 16K and Grape eFP Browser gene chip data in PLEXdb. The expression of *VvGA2oxs*, *VvGA3oxs*, and *VvGA20oxs* gene families in stress was examined by qRT-PCR (Quantitative real-time-PCR). There are 24 *GAoxs* genes identified with the grape genome that can be classified into seven subgroups based on a phylogenetic tree, gene structures, and conserved Motifs in our research. The gene family has higher codon preference, while selectivity is negative selection of codon bias and selective stress was analyzed. The expression profiles indicated that the most of *VvGAox* genes were highly expressed under different time lengths of ABA (Abscisic Acid) treatment, NaCl, PEG and 5 °C. Tissue expression analysis showed that the expression levels of *VvGA2oxs* and *VvGA20oxs* in different tissues at different developmental stages of grapes were relatively higher than that of *VvGA3oxs*. Last but not least, qRT-PCR (Real-time fluorescent quantitative PCR) was used to determine the relative expression of the *GAoxs* gene family under the treatment of GA3 (gibberellin 3) and uniconazole, which can find that some *VvGA2oxs* was upregulated under GA3 treatment. Simultaneously, some *VvGA3oxs* and *VvGA20oxs* were upregulated under uniconazole treatment. In a nutshell, the *GA2ox* gene mainly functions to inactivate biologically active GAs, while *GA20ox* mainly degrades C20 gibberellins, and *GA3ox* is mainly composed of biologically active GAs. The comprehensive analysis of the three classes of *VvGAoxs* would provide a basis for understanding the evolution and function of the *VvGAox* gene family in a grape plant.

## 1. Introduction

Gibberellins (GAs) form a group of tetracyclic diterpenes, and some of them are biologically active. The GA biosynthetic pathway has been elucidated, and its key components have been identified [1]. It serves as hormones in higher plants by controlling diverse growth and developmental processes, such as shoot elongation, leaf expansion and shape, flowering, seed germination, and fruit development [1,2,3,4,5]. Numerous studies have certificated that dwarfism is generally associated with deficiencies in gibberellin (GA) levels or signaling [6,7,8,9,10,11]. GA levels are widely manipulated in agriculture to stimulate fruit growth in seedless grapes, delay fruit senescence in oranges and lemons, increase fruit setting in mandarins, apples and pears, increase stem elongation in sugarcane, or decrease growth in cotton, canola and apple [12]. Furthermore, GAs can prevent post-harvest leaf yellowing in cut flowers, such as lilies and Daffodil [13].

GA2ox, GA3ox, and GA20ox are three key enzymes in GA biosynthesis. These enzymes are members of the 2OG-Fe (II) oxygenase superfamily and independently encoded by different small gene families [13,14]. GA12 and GA53 can be used to synthesize active GA1 and GA4 [5,13], GA2-oxidase is a key enzyme in the degradation process of GAs, and can inactivate the biologically active GAs in plants, thus maintaining biologically active GAs and balancing between the intermediates [15]. However, GA3-oxidase and GA20-oxidase are key enzymes in the synthesis process of GAs, the loss-of-function in GA20 oxidase (GA20ox) and GA3 oxidase (GA3ox) can generate dwarf phenotypes [6,7,8,9,10,11]. Originally described in the runner bean (Phaseolus coccineus) [16], GA 2-oxidases have now been characterized in many plant species from different taxonomic groups and growth habits, including trees [17,18,19,20,21]. The universal occurrence of GA1 and GA4 in plants are functionally active or hormonal forms for growth promotion. GA1 and GA4 combined presence with their biosynthetic precursors and metabolites. Biochemical and molecular data have shown that GA20ox transcript levels are regulated by bioactive GAs through a negative feedback mechanism and in line with diverse environmental factors, indicated that they may control the content of active GAs [22]. Some results have shown that the regulation of GA 20-oxidase transcript levels by auxin is specific to biologically growth-promoting auxins in pea pericarp and the mRNA levels of GA 20-oxidase are differentially regulated by two naturally occurring pea auxins [23]. With overexpressed *ClGA2ox1* or *ClGA2ox3* exhibit dwarf phenotypes in Camellia lipoensis, including reduced growth, delayed flowering, as well as smaller, rounder, and darker green leaves [24]. *GA2ox7* and *GA2ox8* have been identified in Arabidopsis, and shown to be active against C20-GAs, such as GA12 and GA53, but are not capable of using C19-GAs as substrates [22]. Other studies have concluded that C19-GA 2-oxidation is an important GA inactivation pathway regulating the development in Arabidopsis [25].

GA metabolism and signaling pathway are closely controlled by diverse internal and external factors [26,27]. Research found that the biosynthesis of GAs is regulated by plant developmental and environmental stimuli, i.e., bioactive GAs are rapidly reduced when plants are exposed to biotic and abiotic stresses [28,29]. GA3-induced morphological alterations may be related to the control of hormone biosynthesis and signaling, the regulation of transcription factors, the alteration of secondary metabolites, and the stability of redox homeostasis in grape [30]. Seedlessness was induced by the application of GA3 to grape flowers during anthesis severely inhibits pollen germination and pollen tube growth [31,32], possibly because of the biosynthesis of pollen tube inhibitors leading to the generation of unfertilized ovules [33]. Some research has also explored the expression of a regulatory gibberellin oxidation gene under exogenous hormone stimulation and abiotic stress, indicating that gibberellin oxidation gene not only regulates development and growth but also responds to various abiotic stresses [34,35]. In rice, *OsGA2ox5* over-expression was associated with improving resistance to high salinity [36], other findings suggest that *GA20ox2* play an important role in NaCl controlled primary root and root hair growth through its mediation of IAA generation and transport in Arabidopsis [35]. GA biosynthesis inhibitors, such as daminozide (Alar), chlormequat chloride (Cycocel, CCC), paclobutrazol (Benzi), and uniconazole (S3307) are used as growth retardants in the potatoes [37,38]. For example, using uniconazole can change the expression of enzymes involved in hormone biosynthesis and starch metabolic pathways, such as, the significantly increased Abscisic Acid (ABA) and decreased GA modulate the expression of some enzymes involved in starch metabolism, and these enzymes finally resulted in starch accumulation.

Grapevine (*Vitis vinifera* L.) is part of the broadest cultivated and economically significant crops all over the world. The exogenous pre-bloom application of gibberellin 3 (GA3) to grapevine, which is an economically important crop that has long been an important component of the human diet, and is commonly used to induce seedlessness [31,32], establish early ripening [39], and enhance berry size in seedless cultivars [40,41,42]. With the recent report that grape genetic basis for bunch and berry traits could be linked with gibberellin activity related *VvGAI1* gene, which could have a pleiotropic effect and be involved in their genetic regulation, since it is a negative regulator of GAs response [43]. Single nucleotide polymorphisms (SNPs) in the grape genome have been extensively studied, and these findings have led to a better understanding of the genetic diversity of cultivated grapevine [44,45,46]. Although gibberellin oxidase gene has been widely explored [16,30,47,48], a tiny part of *GA2ox*, *GA3ox*, and *GA20ox* gene in grape have been investigated [5,49]. Considering the roles of *GA2ox*, *GA3ox*, and *GA20ox* in response to GA3 during grapevine growth and development, we sought to identify GA2ox, GA3ox, and GA20ox family in grape. In this study, gibberellin oxidase genes were identified and analyzed by bioinformatics method. Using grape abiotic stress and tissue expression data in the GEO (Gene Expression Omnibus) database and screening of the highest expression gene by qRT-PCR (Quantitative real-time-PCR), to explore the function of three gibberellin oxidase genes in the synthesis and degradation of gibberellin, the expression of different tissues in different growth and development stages of grapes, and their response to abiotic stress. This provides a new theoretical basis for grape breeding.

## 2. Materials and Methods

### 2.1. Plant Materials and Treatments

‘Pinot Noir’ tube seedling was used as material for qRT-PCR. The single shoot stem of the seedlings was attached to a solid GS (modified B5 solid medium) and cultured under Light Emitting Diode (LED) white light (light for 16 h and dark for 8 h) in the incubator for 35 days. Subsequently, the seedings were treated by 5, 10, 15, and 20 mg·L^−1^ gibberellin (GA3) and 10 mg·L^−1^ GA3 inhibitor (Uniconazole) for 24 h after cultured for 35 days, and the distilled water was used as a control. All of the materials were collected from grape leaves and frozen in liquid nitrogen and stored at −80 °C for RNA extraction and gene expression analysis.

### 2.2. Identification of Grape Gibberellin Oxidase Genes GA2ox, GA3ox, and GA20ox

All of the sequences were downloaded from four databases: TAIR (Arabidopsis Information Resource) [37], GDR (Apple Information Resource) [50], Rice Genome Annotation Project Database, and Grape Genome Database. Then, a gene of GA oxidase family in Arabidopsis was selected as a seed to blast the three databases. SMART [51] and previous genome annotations are employed to confirm the sequence accuracy and remove genes without the 2OG-Fe (II) oxygenase domain. All of the 24 GA oxidase genes are shown in Table 1, and the information about Arabidopsis, rice, apple, and grape is listed in Appendix A.

The genomic sequences were determined from grape genomes (http://www.genoscope.cns.fr/externe/GenomeBrowser/Vitis/). The gene characteristics, such as PI (Isoelectric Point), MW (molecular weight), GRAVY (grand average of hydropathicity), I.I (instability index), and A.I (aliphatic index), were obtained from EXPASY [52]. The map of the chromosome location with genes was constructed through the online software MapGene2Chrom web v2.

The gene structures (Exon and intron) were analyzed with GSDS v.2.0. The conserved domain of the protein was examined using the MEME online software. In addition, the maximum number of motifs in the conserved domain was placed at 10. 

### 2.3. Phylogenetic Analysis of Systems

The accession numbers of *GA2ox*, *GA3ox*, and *GA20ox* of Arabidopsis thaliana, rice, apple, and grape were aligned using ClustalX v.2.0 (Conway Institute, University College Dublin, Dublin, UK) [50]. A phylogenetic tree was constructed via Molecular Evolutionary Genetics Analysis MEGA 7.0 (Pennsylvania State University, State College, PA, USA) [53] by using the neighbor-joining (NJ) method and adopts the following parameters: the mode adopted “Poisson model,” the gap was set to “Complete deletion,” and the check parameter was bootstrap = 1000 times, random seed.

### 2.4. Codon Usage Bias Analysis and Selective Pressure Analysis

Codon bias refers to the unequal use of synonymous codons for an amino acid [54,55,56,57]. Coding sequences of *VvGA2ox*, *VvGA3ox*, and *VvGA20ox* genes were used to calculate the frequency of optimal codons (FOP), GC content, GC content at the third site of the synonymous codon (GC3s content), relative synonymous codon usage (RSCU), codon adaptation index (CAI), and codon bias index (CBI) with the online software CodonW v.1.4 (http://codonw.sourceforge.net) [58].Correlation analysis between codon composition and preference parameters (A3s, T3s, G3s, C3s, CAI, CBI, FOP, NC, GC3s, GC, L-xym, L-aa, Gravy, and Aromo) was carried out using SPSS v.19.0 (Chicago, IL, USA) statistical software.

The non-synonymous/synonymous (Ka/Ks; ω) value of duplicate gene pairs or triplicate gene groups (between any two genes in one triplicate gene groups) were calculated using DNAsp v.5.0 (University of Barcelona, Barcelona, Spain).

### 2.5. Analysis of the Cis-Acting Element, Subcellular Localization, and Secondary Structure 

The promoter sequences of *GA2ox*, *GA3ox*, and *GA20ox* in grape were downloaded from the grape genome database (2000 bp), and the gene promoter elements were predicted and analyzed using the PlantCARE online site [59,60]. The subcellular localization of GA oxidases was analyzed by WoLF PSORT (https://wolfpsort.hgc.jp/). The secondary structure was identified through NPS@:SOPMA secondary structure prediction.

### 2.6. Acquisition and Analysis of Chip Expression Data in Grape

Expression data were retrieved from microarray platforms (Affymetrix GeneChip 16K Vitis vinifera Genome Array, Affymetrix® Inc., Santa Clara, CA, USA) [61], and the selected data were about ‘Cabernet Sauvignon’ grape under different exogenous ABA treatment conditions (accession numbers: GSE31662 and GSE31664). Abiotic stress data (accession number: GSE31594) were downloaded from the GEO database, and the expression data of GA2ox-, GA3ox-, and GA20ox-related genes were extracted from grapes. Heat maps were drawn using R language. Besides, tissue expression data was obtained from Platforms (GPL13936 NimbleGen 090918 Vitus vinifera exp HX12 [090918_Vitus_exp]) (accession numbers: GSE36128) [62], and the selected data were come from the Grape organs at different stages and the heat map was drawn using TBtools.

### 2.7. RNA Extraction, qRT-PCR and Statistical Analysis

RNA was extracted using a Spectrum Plant Total RNA kit (Sigma St. Louis, MO, USA). Using the extracted total RNA of grape leaves as a template, the reverse strand of complementary DNA (cDNA) was synthesized using the Reverse Transcriptase M-MLV (RNase H-) kit (TaKaRa Biotechnology. Lanzhou, China), and 0.5–2 μg of purified total RNA was reverse transcribed into the first strand cDNA that was used to qRT-PCR. Subsequently, qRT-PCR instrument (Light Cycler 96 Real-Time PCR System, Roche, Basel, Switzerland) was used for the test, and qRT-PCR was performed using TaKaRa SYBR Premix Ex Taq™ II (TaKaRa Biotechnology. Lanzhou, China). Cycling parameters were 95 °C for 30 s, 40 cycles of 95 °C for 5 s, and 60 °C for 30 s. For melting curve analysis, a program including 95 °C for 15 s, followed by a constant increase from 60 °C to 95 °C, was included following the PCR cycles. GAPDH gene (GenBank accession no. CB973647) was used as the internal reference gene, and the primer sequence was shown in Appendix A. The relative expression of the gene was calculated using 2^−ΔΔCT^ method [63]. Samples, which served as cDNA stocks for PCR analysis were stored at −80 °C.

Data quantified from the qRT-PCR of three biological replicates were subjected to two-way ANOVA analysis, followed by Bonferroni’s post-test for data comparison. A *p*-value of less than 0.01 was deemed to represent a significant difference and significant difference analysis using Duncan method.

## 3. Results

### 3.1. Identification of GA2ox, GA3ox, and GA20ox Genes in Grape

A total of 24 candidate genes were obtained in the Grape Genome (12X), which contain 11 GA2ox, 6 GA3ox, and 7 GA20ox, respectively. *VvGA2ox1*–*VvGA2ox11*, *VvGA3ox1*–*VvGA3ox6*, and *VvGA20ox1*–*VvGA20ox7* were appointed on the basis of the order of their gene identification number (ID) (Table 1, Appendix A). All these genes were widely distributed on 13 chromosomes. The largest distribution was located on the 3rd chromosome, and only one gene distributed on the 1st, 2nd, 5th, 7th, 12th, and 16th chromosomes, respectively. It distributes in three genes for the 4th and 19th chromosomes, respectively. Two gene distributions were detected on the 15th and 18th chromosomes, respectively. The CDS coding sequences of *GA2ox*, *GA3ox*, and *GA20ox* in grape ranged from 825 bp (*VvGA3ox1*) to 2037 bp (*VvGA3ox4*), amino acid sequence lengths ranged from 275–678. The molecular weight of these three types of gibberellin oxidase with large differences that between 31.27–75.98 kD. The largest difference in molecular weight is the GA3ox class with a difference of 44.71 kD. The molecular weight difference of the GA20ox class is 13.58 kD, and the smallest difference in molecular weight of GA2ox class with a difference of 12.48 kD. Three types of gibberellin oxidase proteins had hydrophilic values ranging from −0.05 to −0.54 and all of them were hydrophilic proteins. Predicted values of the isoelectric points of these three types were among 5.14–8.22, with only one basic protein (GA2ox2), one neutral (GA2ox8), and the remaining proteins were acidic. Furthermore, 72.73% (8) and 66.67% (4) of GA2ox and GA3ox proteins had an instability index of >40, respectively. However, 42.86% (3) of the GA20ox proteins had an instability index of >40, indicating that GA2ox and GA3ox oxidases were more stable than GA20ox oxidase.

### 3.2. Structural Analysis of GA2ox, GA3ox, and GA20ox Genes

The phylogenetic tree was built using GA2ox, GA3ox, and GA20ox protein sequences in grape, and gene structure analysis was performed (Figure 1). The amino acid sequence comparisons revealed that grape GAoxs had a conserved Fe^2+^ 2-oxoglutarate-dependent dioxygenase (2-ODD) domain in each ODD (Appendix A). The number of exons of the 24 gibberellin oxidase genes, including *VvGA3ox2* and *VvGA3ox3*, was between 2 and 6. *VvGA3ox5* and *VvGA3ox6* had only 2 exons, and *VvGA3ox4* had 6 exons. The closer the genetic relationship was, the more similar the genetic structure between them would be, such as those between *VvGA20ox2* and *VvGA20ox6* and between *VvGA2ox9* and *VvGA2ox10*, which contained the same number of exons, and have the same structure of genes. Three types of gibberellin oxidase genes were mostly within 3 kb, and the individual genes *VvGA2ox3* and *VvGA3ox4* reached about 6 kb, which might be related to their special features.

The phylogenetic tree was constructed using the GA2ox, GA3ox, and GA20ox proteins of Arabidopsis thaliana (At), Oryza sativa (Os), Malus domestica (Md), and Vitis vinifera (Vv) (Figure 2). In line with the evolutionary relationship and according to the Huang et al., 2015 research [50], they were divided into seven subgroups, namely GAox-A, GAox-B, GAox-C, GA20ox, GA3ox, C19 GA2ox and C20 GA2ox subgroups. The evolutionary relationship showed that the distribution of GA2ox, GA3ox, and GA20ox in each species was similar and evenly distributed in the seven subfamilies. We can classify VvGA2ox3, 4, 5 to C20 GA2ox subgroup, according to C20 GA2ox subfamily members of Arabidopsis that containing AtGA2ox7, 8 and OsaGA2ox5. VvGA3ox2, 3 and 6 can be classified in the GA3ox subfamily. VvGA2ox1, 2, 7, can be classified in the C19 GA2ox subfamily. And VvGA20ox1, 2, 4, 5 and 6 were classified in the GA20ox subfamily. In addition, VvGA20ox7, VvGA2ox9 and VvGA2ox10 can be classified to GAox-A. VvGA3ox1, VvGA2ox8 and VvGA2ox11 can be classified to GAox-B. VvGA20ox3, VvGA3ox5, VvGA3ox4 and VvGA2ox6 can be classified to GAox-C. Overall, the number of GA20ox and GA2ox was greater than that of GA3ox, indicating that GA20ox and GA2ox also experienced a dynamic evolutionary route and leading to functional redundancy.

The conservative motif analysis of different subgroups of protein sequences was performed using MEME to obtain 10 conserved motifs (Figure 3) and named motif 1 to motif 10 (Appendix A). Motifs 1, 2, and 4 were conserved sequences shared by all of the genes (Appendix A). Among them, motif 3 is unique to GA2ox and GA3ox. motifs 5, 6, and 8 are unique to GA2ox and GA20ox. motifs 7 and 9 are in GA2ox and GA3ox. motif 10 was only present in GA2ox9, 10, and GA20ox7. Six pairs of genes with homology greater than 90 were VvGA20ox2/VvGA20ox6, VvGA2ox3/VvGA2ox5, VvGA2ox9/VvGA2ox10, VvGA3ox2/VvGA3ox3, VvGA2ox8/VvGA2ox11, and VvGA2ox1/VvGA2ox2. The conserved sequence and protein lengths of each pair of proteins were basic and identical.

We identified ten triplicated gene groups in the grape *VvGA2ox*, *VvGA3ox*, and *VvGA20ox* gene family, such as the *VvGA3ox2*–*VvGA3ox3*–*VvGA3ox6* group. In addition, *VvGA3ox2*, *VvGA3ox3* and *VvGA3ox6* were collinear, showing that the triplication may derive from chromosome segmental triplication or a large-scale triplication event. All the triplicated gene groups in the grape *VvGA2ox*, *VvGA3ox*, and *VvGA20ox* family underwent purifying selection (Appendix A). Based upon our chromosomal location, gene structure and motif analysis, we found that the genes may be functionally conserved within a triplicated gene group such as the *VvGA3ox2*–*VvGA3ox3*–*VvGA3ox6*, *VvGA2ox3*–*VvGA2ox5*–*VvGA2ox4* and *VvGA20ox4*–*VvGA20ox5*–*VvGA20ox1* groups. The motifs of one *VvGA2ox*, *VvGA3ox*, and *VvGA20ox* family member are also similar to the other members in the triplicated same gene group (Figure 1, Figure 2 and Figure 3).

### 3.3. Codon Usage Bias Analysis 

Analysis of codon usage parameters of *VvGA2ox*, *VvGA3ox*, and *VvGA20ox* gene families was listed in Appendix A. Among these parameters, CAI, CBI, and NC are usually predicting gene expression levels. Generally, CAI and CBI are positively correlated, with values ranging from 0 to 1 and the closer a value is to 1; the stronger the codon preference, the higher the gene expression level, and the negative correlation to the number of effect codons (NC). NC value is generally from 20 to 61. 20 indicates that the synonymous codon bias is larger, and the closer to 61, the smaller the synonym codon bias. Highly expressed genes have a large degree of codon preference, with large CAI and CBI values and small NC values, low expression genes contain more types of rare codons, so the preference is low, CAI and CBI values are smaller and NC values are higher. Its value will be influenced by the amino acid composition of the gene and the length of the gene. From the data point of view, NC values of the *VvGA2ox*, *VvGA3ox*, and *VvGA20ox* gene families ranged from 45.88–57.27, and the expression level was average. We can find that the CAI and CBI values of *VvGA2ox7* are quite high, while the NC values are relatively low, and suspected the expression level is relatively high. It is generally accepted that the two factors of GC amount and GC3s have an important influence on the codon usage of the gene. In these three gibberellin oxidase gene sequences, the amount of 21 gibberellin oxidase gene GC was less than 0.5, indicated that the grape gibberellin oxidase gene had no discernible preference for GC. The GC3s value was greater than 0.5, accounting for 41.7%, indicated that most of the grape gibberellin oxidase genes prefer codons ending in A/T.

A correlation analysis was conducted to further understand the influences of codon bias on gene properties (Appendix A). On the basis of the correlation analysis, we can find that FOP and CBI, CAI is an extremely significant correlation, simultaneously, the content of GC and GC3s is a significant correlation, through these consequences, we can infer the gibberellin oxidase gene is mainly derived from the stress of mutation. Gravy values were negative and they were significantly negatively correlated with GC3s (*p* < 0.01), so gibberellin oxidase proteins were all hydrophobic proteins. The GC (G + C content) in the grape was significantly positively correlated with G3s (the frequency of occurrence of the corresponding base in the third position of the synonymous codon) and GC3s (the G + C content in the third position of the codon) (*p* < 0.01), which is significantly negatively correlated with T3s and A3s (*p* < 0.01), FOP (optimal codon usage frequency) and CBI (codon preference index), CAI (codon adaptation). 

The indices were all significantly positively correlated (*p* < 0.01). GC and GC3s were significantly negatively correlated with T3s and A3s (*p* < 0.01), and GC, GC3s and NC (codon effective numbers) were substantially negative. Correlation (*p* < 0.05) indicates that base composition has an important influence on codon preference and gene expression level. The level of gene expression and codon usage preference is affected by the synonym codon the effect of the base in the third position. The more codon-preferred (lower NC values) genes, the more preferred they are to use optimal codons (higher FOP values) and higher G + C content, especially at G/C (larger GC3s). 

RSCU is the relationship between the actual number of codons corresponding to an amino acid and the number of theoretical applications. It can intuitively reflect the degree of preference of codon usage, regardless of gene length and amino acid abundance. When the actual number of applications is the same as the number of theoretical applications, RSCU = 1, there is no preference for codons; when RSCU > 1, it indicates that the codon appears more frequently than other synonymous codons. Preferred codons; when RSCU < 1, the relative frequency of codons is low, which is a codon that is less frequently used by genes. By comparison (Appendix A), in *VvGA2ox*, there are 23 the codon of RSCU > 1, 20 VvGA3ox, RSCU > 1, 21 VvGA20ox, RSCU > 1, Among them, the RSCU values of 17 codons in the grape *GA2ox*, *GA3ox*, and *GA20ox* were all greater than 1, indicating that these codons are preferred codons for the gene family. These codons are the most commonly used codons for the gibberellin oxidase gene, the optimal codon.

### 3.4. Subcellular Localization and Secondary Structure Analysis 

The results of subcellular localization (Appendix A) showed that the gibberellin oxidase gene family was mainly expressed in the chloroplast, the cytoplasm, and the nucleus; indicating that the gene family mainly existed in organs with strong photosynthesis and respiratory metabolism. Among them, VvGA2ox1, 4, VvGA20ox1, and 5 were not noted in the cytoplasm. VvGA2ox3, 9, VvGA20ox3, and 7 were not expressed in the chloroplasts, and the remaining genes were written in the cytoplasm. VvGA2ox5 and 7 were not expressed in the nucleus, but the other genes were written in the nucleus. Only four genes VvGA3ox2, 4, VvGA20ox5, 6 expressing in the mitochondria, and other genes were not noted. Only three genes, namely, VvGA2ox10, VvGA3ox4, and VvGA20ox5, were associated with the peroxisome. VvGA20ox1 and 2 were written in vacuoles. VvGA2ox1 and 2 were expressed in the Golgi. VvGA2ox10, 11, VvGA3ox1, 5, VvGA20ox2, 3, 4, and 6 were noted in the cytoskeleton, and these genes might be associated with cell wall formation. 

The secondary structure of the grape gibberellin oxidase gene family encodes proteins mainly consisting of α-helix, β-turn, and irregular curl. Secondary structure analysis indicated that the 24 proteins encoded by the Gibberellin oxidase gene family were mainly α-helix and irregular coils (Appendix A).

### 3.5. Gene Chip Expression Profile Analysis

These three kinds of genes were removed by downloading the gene expression data of ABA and abiotic stress from the GEO database to predict the expression of gibberellin oxidase gene in response to ABA and abiotic stress. The data corresponding to the grapes were tested by heat map (Figure 4 and Figure 5). The expression of gibberellin oxidase gene in response to different periods of ABA stress significantly differed (Figure 4). With the prolongation of stress time, the genes with continuously upregulated expression levels were *VvGA2ox9*, *10*, and *VvGA20ox7*. *VvGA2ox1*, *8*, *9*, *10*, and *11* were continuously upregulated under ABA stress for 3, 10, and 14 d. Only the expression levels of *VvGA2ox9*, *10*, and *VvGA20ox7* were upregulated under ABA stress for 28 d, indicating that ABA was closely related to *VvGA2ox9*, *10*, and *VvGA20ox7*.

In response to different abiotic stresses, the expression of gibberellin oxidase gene was further significantly different (Figure 5). *VvGA2ox2*, *3*, *5*, *6*, *VvGA20ox2*, *4*, *6*, *7*, *VvGA3ox1*, *4*, and *5* were upregulated in response to salt stress. The expression of *VvGA2ox7*, *VvGA20ox2*, and *VvGA3ox4* was stable. *VvGA20ox1* also maintained a low expression level. The expression levels of *VvGA2ox2*, *3*, *4*, *5*, *6*, *7*, *9*, *10*, *VvGA20ox2*, *3*, *5*, *6*, *7*, *VvGA3ox1*, *4*, and *5* were significantly upregulated under PEG stress for 24 h. The expression levels of *VvGA2ox2*, *3*, *4*, *5*, *VvGA20ox3*, and *VvGA3ox5* were upregulated. The expression levels of *VvGA2ox1*, *8*, *11*, *VvGA20ox1*, *6*, *7*, and *VvGA3ox6* were downregulated under 24 h of PEG stress. The expression levels of *VvGA2ox1*, *4*, *6*, *8*, *11*, *VvGA20ox1*, *3*, *5*, *VvGA3ox1*, 4, and 6 were upregulated in response to low-temperature stress at 5 °C. The expression levels of *VvGA2ox1*, *8*, *11*, *VvGA20ox1*, and *3* under low-temperature stress at 5 °C for 8 h were upregulated. These findings suggested that these upregulated genes had a certain role in the cold resistance of grape.

Fourteen grape gibberellin oxidase genes were found in the Grape eFP Browser database, and the expression of these genes in different tissues of different growth stages of wine grapes was visualized using the TBtool (Figure 6). Overall, the expression levels of *VvGA2ox* in different tissues were significantly higher than those of *VvGA3ox* and *VvGA20ox*. From a local perspective, *VvGA2ox1* was expressed in other tissues except for the low expression levels in roots and seeds. *VvGA2ox2* is mainly in Stamen, BerryPericarp-FS, BerryPericarp-PFS, Bud-SPericarp-V, BerryPericarp-MR, BerryPericarp-R, Bud-S, Bud-B, Bud-L, BerryFlesh-PFS, BerryFlesh-V, BerryFlesh-MR, BerryFlesh-R, Inflorescence-Y, Inflorescence-WD, Flower-FB, Flower-F, Rachis-FS, Rachis-PFS, Rachis-V, Rachis-MR, Rachis-R, BerrySkin-PFS, BerrySkin-V, BerrySkin. The expression in -MR, BerrySkin-R, Steam-G, Tendril-Y, Tendril-WD is relatively high. *VvGA2ox6* is mainly expressed in BerryPericarp-PFS, BerryPericarp-MR, Bud-S, Bud-AB, Bud-W, Inflorescence-Y, Root, Leaf-Y, Seeding, BerrySkin-MR, Steam-W. *VvGA2ox7* is mainly in BerryFlesh-PHWI, BerryFlesh-PHWII, BerryFlesh-PHWIII, BerryPericarp-PHWI, BerryPericarp-PHWII, BerryPericarp-PHWIII, Seed-V, Seed-MR, Seed-FS, Seed-PFS, BerrySkin-PHWI, BerrySkin-PHWII The expression level in BerrySkin-PHWIII is relatively high. *VvGA2ox9* is mainly expressed in Bud-W, *VvGA2ox10* is mainly expressed in Root, and *VvGA2ox11* is mainly expressed in BerryFlesh-PFS, Leaf-S, Seed-V, Seed-FS, Seed-PFS. *VvGA3ox3* and *VvGA3ox6* are mainly expressed in Pollen, and *VvGA3ox2* is mainly expressed in Bud-B, Inflorescence-Y, Seed-FS, Seeding, Tendril-Y, Tendril-WD, Tendril-FS. *VvGA3ox5* is mainly expressed in Bud-B and BerrySkin-MR. *VvGA20ox4* is mainly expressed in Stamen, Flower-FB, Flower-F, Carpel, Petal, Pollen, Seed-PFS, *VvGA20ox5* is highly expressed in Seed-FS, Seed-PFS, and *VvGA20ox7* is different in wine grapes. The expression levels in different tissues during development are relatively high.

### 3.6. Gene Cis-Element Analysis of GA2ox, GA3ox, and GA20ox 

PlantCARE was used to predict the upstream promoter elements of the gene and analyze the number of cis-acting elements related to hormones, meristems, adversity, and circadian rhythm to further explore the functions of *GA2ox*, *GA3ox*, and *GA20ox* (Figure 7), and the main cis-element selected in the Appendix A. On the promoters of these three types of gibberellin oxidase, stress-related action elements were ubiquitous and relatively abundant. Hormone-related action elements, such as methyl jasmonate, gibberellin, and abscisic acid, were compared. In general, the amounts of auxin and salicylic acid were relatively small, and only six genes, namely, *VvGA2ox2*, *VvGA20ox1*, *4*, *5*, *VvGA3ox1*, and *6*, were associated with meristematic tissues, and four genes, namely, *VvGA2ox2*, *6*, *VvGA3ox6*, and *VvGA20ox1*, were related to circadian rhythm. Although all of the three genes are owned by the gibberellin oxidase gene, some of the gibberellin oxidase gene promoters contain gibberellin-related elements, such as *VvGA2ox2*, *3*, *5*, *9*, *VvGA3ox6*, *VvGA20ox1*, and *7.* On the promoters of *GA20ox* genes, only *VvGA20ox1* and *7* did not contain gibberellin-related action elements, and others contained gibberellin-related action elements. Many genes contain abiotic stress elements, such as low temperature, drought, saline and other stress response components, while six genes do not get this cis-acting element. Specifically, the promoters of different genes contained the same species or even a number of similar or identical elements, and the similarity of their functions should need to be further verified.

### 3.7. Effects of GA3 and Uniconazole Influence the Expression of GA2ox, GA3ox, and GA20ox

‘Pinot Noir’ seedlings grown in Gansu Agricultural University, Lanzhou, China for 35 d were treated with 5, 10, 15, and 20 mg·L^−1^ GA3 and 10 mg·L^−1^ uniconazole. After 24 h of treatment, RNA was extracted separately, and the expression levels of *GA2ox*, *GA3ox*, and *GA20ox* were detected through qRT-PCR (Figure 8, Appendix A).

Under different concentrations of gibberellin treatment, the expression levels of *GA2ox* were significantly upregulated, and the expression levels of *GA3ox* and *GA20ox* were downregulated. The expression levels of *GA3ox* and *GA20ox* were obviously upregulated when the samples were treated with 10 mg·L^−1^ uniconazole. After 5 mg·L^−1^ gibberellin 3 (GA3) treatment, the upregulated genes were *VvGA2ox1*, *3*, *6, 7*, *9*, *10*, *11*, *VvG3ox4*, and *VvGA20ox7*. Among them, *VvGA2ox7* was the highest, which relative expression was reached to 46.66 folds to compare with the control. Followed by the expression of *VvGA2ox1* and *11*, whose expression levels were 4.34 and 4.69 time that of the control, respectively. Under the treatments of the 10 mg·L^−1^ GA3, *VvGA2ox5*, *8*, *VvGA3ox1*, *VvGA20ox3*, *5*, and *6* were upregulated. The highest expression was observed in *VvGA20ox5*, which was 3.45 times more than control. Under the treatments of the 15 mg·L^−1^ GA3 was administered, the expression levels of *VvGA2ox1*, *11*, *VvGA3ox4*, and *5* were upregulated. The gene with the highest expression was *VvGA2ox11*, which was 4.28 time of the control. After treatment with 20 mg·L^−1^ GA3 was administered, *VvGA2ox1*, *7*, *11*, *VvGA3ox5*, and *VvGA20ox5* were 0.90, 1.78, 1.69, 1.13, and 1.79 time of the control, respectively. After treatment with 10 mg·L^−1^ the gibberellin inhibitor uniconazole was given, the upregulated genes were *VvGA2ox3*, *6*, *10*, *VvGA3ox2*, *4*, *5*, *VvGA20ox1*, *2*, *3*, *4*, *6*, and *7*. Among them, *VvGA20ox7* expression level was 2.58 times of the control.

## 4. Discussion

Gibberellins are associated with several aspects of growth and development, including seed maturation, stem elongation, floral induction, germination, pollen germination, and pollen tube growth [2,12,13,33,64]. Nevertheless, few studies have been conducted on *GA2ox*, *GA3ox*, and *GA20ox* in response to hormonal and abiotic stresses [34,38]. Therefore, we sought to study the characteristics of this gene family and the expression of gibberellin oxidase genes under gibberellin and uniconazole treatment was quantitatively analyzed in qRT-PCR to verify the mechanism of these genes in grape and analysis the expression of gene chip expression profile is related to ABA, abiotic stress and different tissues of grape.

In this study, we have identified and analyzed of *GA2ox*, *GA3ox*, and *GA20ox* gene. The effects of these three gibberellin oxidase genes on gibberellin synthesis has been explored and the role of the gene family in grapes has been examined. In this experiment, 24 gibberellin oxidase genes from the grape genome have been identified by bioinformatics analysis, but the number is different from other plants, such as 16 members in Arabidopsis thaliana [38], 21 members in rice [65,66], 24 members in soybean [37], and 18 members in cucumber [67]. The gibberellin oxidase family is large, which may be linked to genome size. VvGA2ox, VvGA3ox, and VvGA20ox were widely distributed on 13 chromosomes except chromosomes 6, 8, 11, 13, 14, and 17. All of them were allocated, and each chromosome had one to four genes possibly because GA was involved in various life activities of plants (seed germination, stem elongation, fruit development, etc.). Specific motifs in amino acid sequences are vital regions related to function. As subfamilies from 2OG-Fe (II) oxygenase superfamily, GA oxidases include the C-terminal of prolyl 4-hydroxylase α subunit, and the parts of the active site comprise an alpha2 beta2 complex with an α subunit [49,68,69]. The analysis of gene structure and conserved motifs revealed similar genetic structures with comparable genetic evolutionary relationships. The conservative motif analysis of grapes found that motif 3 was unique to GA2ox and GA3ox. Motif 6 and motif 8 were unique to GA2ox and GA20ox. Similar reports have also described the conserved motifs in Arabidopsis and cucumber species [37]. Similarly, different types of genes contained specific motifs possibly because of different types of gibberellin oxidases. The precise relationship should be further studied.

The evolutionary properties of the gibberellin oxidase gene family in Arabidopsis, rice, apple, and grape showed that functionally different GA2ox, GA3ox, and GA20ox clusters were distributed in separate subfamilies. OsGA20ox3, 5, 6, 7, OsGA2ox11, OsGA3ox1, 2, and 5 were found in rice [37,49]. MdGA20ox4, MdGA3ox1, MdGA2ox3, and MdGA2ox11 were found in apple [20]. AtGA2ox4, AtGA3ox1, 2, and 4 were found in Arabidopsis [37]. VvGA2ox6, 9, 10, VvGA3ox4, 5, VvGA20ox3, and 7 were detected in grape [5]. Combining the results of Han et al., and Huang et al., we divided the three gibberellin oxidase gene families into seven subfamilies: GAox-A, GAox-B, GAox-C, GA20ox, GA3ox, C19 GA2ox and C20 GA2ox subgroups [37,49]., and an evolutionary relationship likely exists between the gibberellin oxidase genes of rice, apple, Arabidopsis and grape [37,70,71]. The tandem repeat of the gene combined with the gene structure, evolutionary analysis, motif analysis found that the genes distributed in the same subfamily have the same structure and motif, and the gene duplication and triplication are identical in structure and motif.

The results of the promoter cis-acting element analysis showed that GA2ox, GA3ox, and GA20ox contained some cis-acting elements related to hormones, meristems, stress, and circadian rhythm regulation. These genes were also hit by GA regulation. Jasmonic acid, IAA, SA, ABA, and stress regulation are consistent with previous studies [72]. These three genes belonged to the gibberellin oxidase gene, but some genes, such as VvGA2ox2, 3, 5, 9, VvGA3ox6, VvGA20ox1, and 7, and other gene promoters do not contain gibberellin-related elements, possibly regulation of endogenous GA content changes by other factors on the GA signaling pathway to regulate GA levels in plants. In total, 24 genes had hormone-, meristem-, stress-, and circadian rhythm-related elements on the upstream promoter. The expression data of these three genes under ABA and abiotic stress were downloaded and analyzed by the GEO database. The results showed that these three genes could respond to different duration of ABA and various abiotic stress conditions [36]. Besides, the expression analysis of different tissues in different developmental stages of grape showed that *GA2ox7* was mainly expressed in seeds and fruits in C19 GA2ox subfamily. The expression of *GA2ox1*, *2* was basically the same, mainly expressed in fruits, buds, stamens, flowers, core skin and other tissues. *GA20ox7* was highly expressed in roots, stems, leaves, flowers, buds, fruits and other tissues in the GA20ox subfamily. Compared with *GA2ox* and *GA20ox*, *GA3ox* is mainly expressed in stems, buds, inflorescences and pollen. We combined the codon preference analysis, grape gene chip expression and fluorescence quantitative analysis found that the three analysis results have similarity, that is, the expression level of *VvGA2ox7* is relatively high.

The genes of the gibberellin oxidase gene in grape were treated with different concentrations of GA and uniconazole by qRT-PCR to investigate the relationship between *GA2ox*, *GA3ox*, and *GA20ox* in grapes, GA, and uniconazole. The results showed that the expression of *GA2ox* was upregulated in these genes after GA3 treatment, because the GA2-oxidase is a key enzyme in the degradation process of GA, which can inactivate the biologically active GAs and their precursors and other intermediates in plants, thus maintaining biologically active GAs in plants and balance between intermediates [15]. A previous analysis indicated that feedback regulation controls the concentrations of active GAs in higher plants [13]. In most plants, GA20ox and GA3ox, whose products oxidize the penultimate and final steps, respectively, and the production of bioactive GAs (GA1 and GA4) are downregulated by applied exogenous GA [13]. In contrast, the genes encoding GA2ox, which convert active GAs to inactive catabolites, are upregulated by GA treatment [16]. Our findings are similar to previous studies, our results in accordance with others consequences that indicated the GA2ox gene mainly functions to inactivate biologically active GAs, while GA20ox mainly degrades C20 gibberellins, and GA3ox is mainly composed of biologically active GAs [5,13]. In a nutshell, our study provides new insights into the functional divergence of the GAox family in grape and implicates the potential roles of positive selection in the evolution of *GAox* genes, which deserves further investigation.

## 5. Conclusions

Twenty-four gibberellin oxidase genes, including *GA2ox*, *GA3ox*, and *GA20ox*, were found in grape, which can be subdivided into seven subfamilies. *GA2ox*, *GA3ox*, and *GA20ox* were associated with abiotic stress expression and response to the expression of various exogenous hormones, they have different expressions in different organizations. The *GA2ox* gene mainly functions to inactivate biologically active GAs, while *GA20ox* mainly degrades C20 gibberellins, and *GA3ox* is mainly composed of biologically active GAs. Our findings provided a basis for conducting in-depth mechanistic studies on the distinct biological characteristics and adaptability of the grape in harsh environments.

## Figures and Tables

**Figure 1 genes-10-00680-f001:**
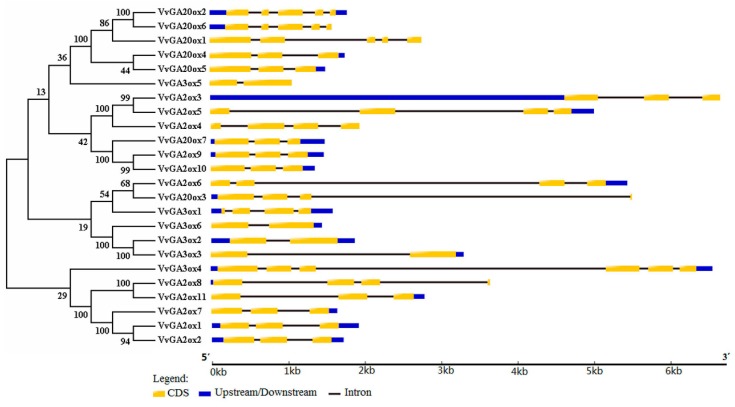
Phylogenetic tree and gene structure analysis of *GA2ox*, *GA3ox*, and *GA20ox* gene family in grape. The amino acid sequences of GA2ox, GA3ox, and GA20ox proteins were aligned with ClustalX, and a phylogenetic tree was constructed using the Neighbor-joining method (NJ) method in MEGA v.7.0. Each node was represented by a number that indicated the bootstrap value for 1000 replicates. The scale bar represented 0.1 substitution per sequence position (left). The right side illustrates the exon–intron organization of the corresponding *GA2ox*, *GA3ox*, and *GA20ox* genes. Exon CDS, Upstream/Downstream, and intron were denoted by the yellow boxes, blue boxes, and black lines, respectively. The scale bar represented 1 kb (right).

**Figure 2 genes-10-00680-f002:**
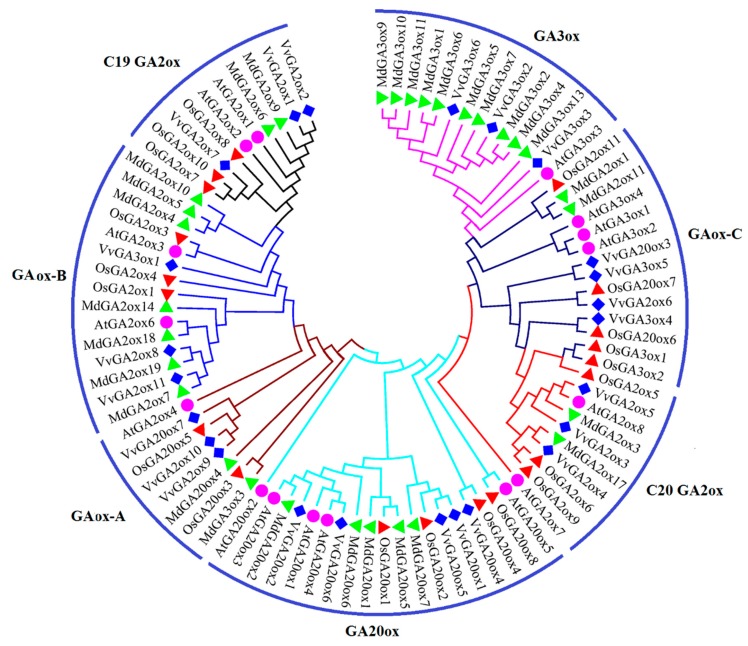
Phylogenetic analysis of *GA2oxs*, *GA3oxs*, and *GA20oxs* in *Arabidopsis* (At), rice (Os), *Malus* (Md), and grape (Vv). The amino acid sequences of GA2oxs, GA3oxs, and GA20oxs proteins were aligned with ClustalX, and a phylogenetic tree was constructed using the NJ method in MEGA v.7.0. Each node was represented by a number that indicated the bootstrap value for 1000 replicates. The subgroups were marked by a colorful background.

**Figure 3 genes-10-00680-f003:**
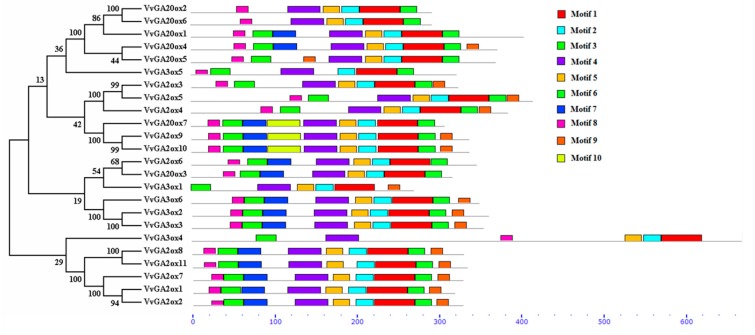
Schematic of the amino acid Motifs of GA2ox, GA3ox, and GA20ox proteins in grape.

**Figure 4 genes-10-00680-f004:**
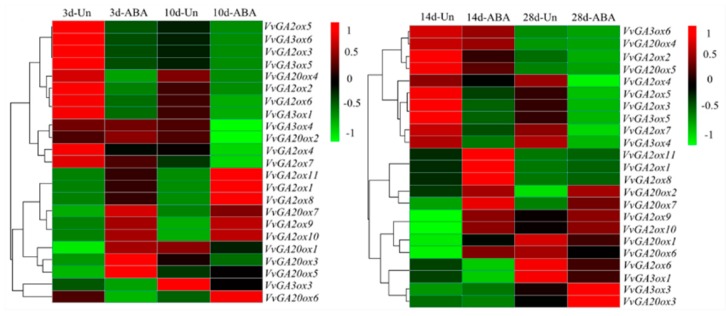
Hierarchical clustering of the expression profiles of 23 *VvGAox* genes subjected to ABA treatments for 3, 10, 14, and 28 days in grape. In the graph, 3d-Un represent untreated the grape fruit just as control, 3d-ABA represent have been treated grape fruit with ABA, Simultaneously, 10d-Un, 14d-Un, 28d-Un, representing untreated grape fruit at 10, 14, and 28 days, otherwise, 10d-ABA, 14d-ABA, 28d-ABA, representing have been treated the grape fruit at 10, 14, and 28 days. Red or green shading represented the upregulated or downregulated expression level, respectively. The scale denoted the relative expression level.

**Figure 5 genes-10-00680-f005:**
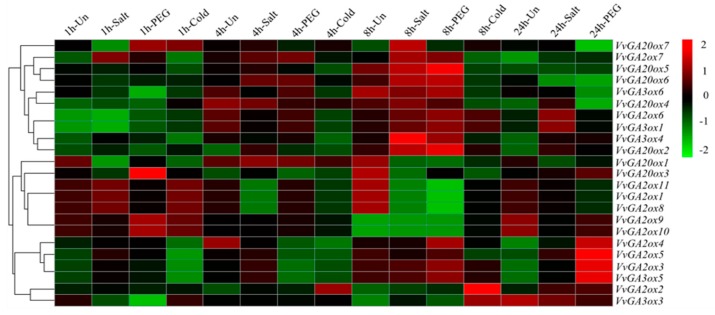
Hierarchical clustering of the expression profiles of 23 *VvGAoxs* genes at different abiotic stress experiments in grape. Abiotic stress experiments: salt, PEG, and cold. Heatmap experiments were performed with GeneChip microarrays, which were from Affymetrix GeneChip 16K with short-term abiotic stress‘Cabernet Sauvignon.’ Red or green shading represented the upregulated or downregulated expression level, respectively. The scale denoted the relative expression level.

**Figure 6 genes-10-00680-f006:**
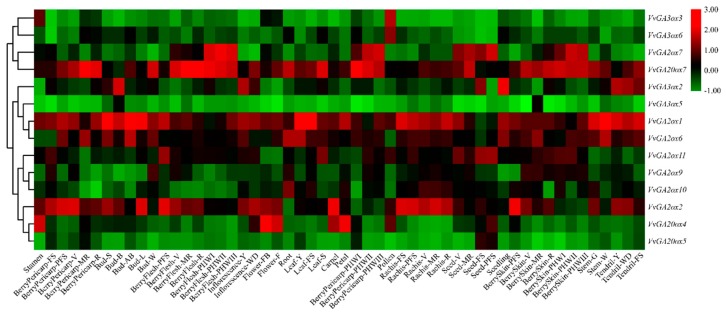
Hierarchical clustering of the expression profiles of 14 *VvGAoxs* genes at different organizations experiments in grape. Heatmap experiments were performed with GeneChip microarrays, which were from Grape eFP Browser in grape. Red or green shading represented the upregulated or downregulated expression level, respectively. The scale denoted the relative expression level. **Note:** Stamen, pool of stamens from undisclosed flowers at 10% and 50% open flowers; BerryPericarp-FS, berry pericarp fruit set; BerryPericarp-PFS, berry pericarp post-fruit set; BerryPericarp-V, berry pericarp véraison; BerryPericarp-MR, berry pericarp mid-ripening; BerryPericarp-R, berry pericarp ripening; Bud-S, bud swell; Bud-B, bud burst (green tip); Bud-AB, bud after-burst (rosette of leaf tips visible); Bud-L, latent bud; Bud-W, winter bud; BerryFlesh-PFS, berry flesh post fruit set; BerryFlesh-V, berry flesh véraison; BerryFlesh-MR, berry flesh mid-ripening; BerryFlesh-R, berry flesh ripening; BerryFlesh-PHWI, berry flesh post-harvest withering I (1st month); BerryFlesh-PHWII, berry flesh post-harvest withering II (2nd month); BerryFlesh-PHWIII, berry flesh post-harvest withering III (3rd month); Inflorescence-Y, young inflorescence (single flower in compact groups); Inflorescence-WD, well developed inflorescence (single flower separated); Flower-FB, flowering begins (10% caps off); Flower-F, flowering (50% caps off); Root, root in vitro cultivation; Leaf-Y, young leaf (pool of leaves from shoot of 5 leaves); Leaf-FS, mature leaf (pool of leaves from shoot at fruit set); Leaf-S, senescencing leaf (pool of leaves at the beginning of leaf fall); Carpel, pool of carpels from undisclosed flowers at 10% and 50% open flowers; Petal, pool of petals from undisclosed flowers at 10% and 50% open flowers; BerryPericarp-PHWI, berry pericarp post-harvest withering I (1st month); BerryPericarp-PHWII, berry pericarp post-harvest withering II (2nd month); BerryPericarp-PHWIII, berry pericarp post-harvest withering III (3rd month); Pollen, pollen from disclosed flowers at more than 50% open flowers; Rachis-FS, rachis fruit set; Rachis-PFS, rachis post-fruit set; Rachis-V, rachis véraison; Rachis-MR, rachis mid-ripening; Rachis-R, rachis ripening; Seed-V, seed véraison; Seed-MR, seed mid-ripening; Seed-FS, seed fruit set; Seed-PFS, seed post-fruit set; Seedling, seedling pool of 3 developmental stages; BerrySkin-PFS, berry skin post-fruit set; BerrySkin-V, berry skin véraison; BerrySkin-MR, berry skin mid-ripening; BerrySkin-R, berry skin ripening; BerrySkin-PHWI, berry skin post-harvest withering I (1st month); BerrySkin-PHWII, berry skin post-harvest withering II (2nd month); BerrySkin-PHWIII, berry skin post-harvest withering III (3rd month); Stem-G, green stem; Stem-W, woody stem; Tendril-Y, young tendril (pool of tendrils from shoot of 7 leaves); Tendril-WD, well developed tendril (pool of tendrils from shoot of 12 leaves); Tendril-FS, mature tendril (pool of tendrls at fruit set).

**Figure 7 genes-10-00680-f007:**
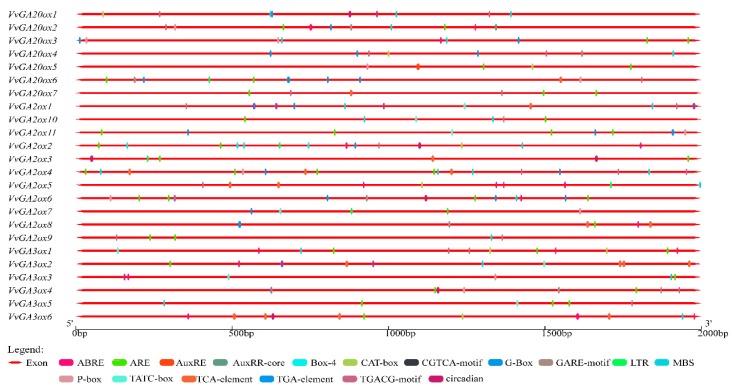
Number of cis-acting elements in GA2ox, GA3ox and GA20ox promoter of grape. **Note**: ABRE, abscisic acid responsiveness. ARE, regulatory element essential for the anaerobic induction. AuxRE, auxin-responsive element. AuxRR-core, regulatory element involved in auxin responsiveness. Box-4, part of a conserved DNA module involved in light responsiveness. CAT-box, cis-acting regulatory element related to meristem expression. CGTCA-Motif, cis-acting regulatory element involved in the MeJA-responsiveness. G-Box, cis-acting regulatory element involved in light responsiveness. GARE-Motif, gibberellin-responsive element. LTR, cis-acting element involved in low-temperature responsiveness.MBS, MYB binding site involved in drought-inducibility. P-box, gibberellin-responsive element. TATC-box, cis-acting element involved in gibberellin-responsiveness. TCA-element, cis-acting element involved in salicylic acid responsiveness. TGA-element, auxin-responsive element. TGACG-Motif, cis-acting regulatory element involved in the MeJA-responsiveness. Circadian, cis-acting regulatory element involved in circadian control.

**Figure 8 genes-10-00680-f008:**
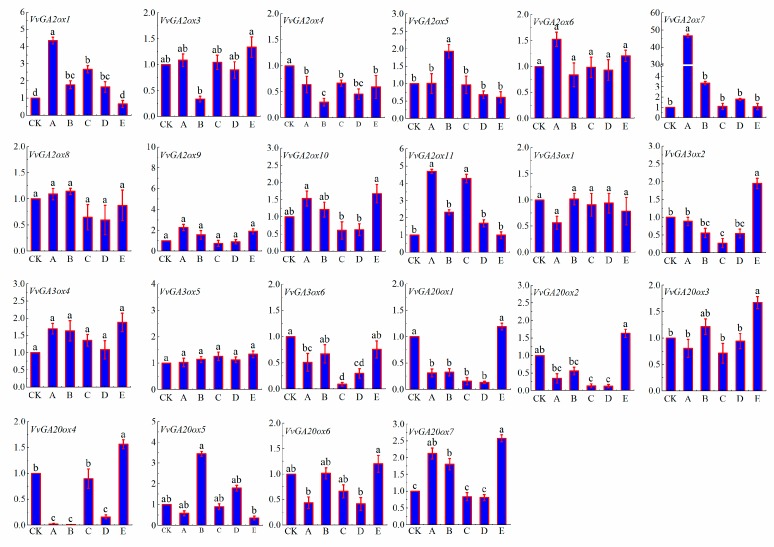
Expression levels of VvGA2ox, VvGA3ox, and VvGA20ox in grape leaves after 24 h under different treatments with CK, contrast check; A, 5 mg·L-1 gibberellin 3 (GA3) ; B, 10 mg·L-1 GA3 ; C, 15 mg·L-1 GA3 ; D, 20 mg·L-1 GA3 ; and E, 10 mg·L-1 uniconazole. The specimens were analyzed through real time PCR. GAPDH (CB973647) was detected as an internal reference gene. Gene expression was normalized to the control unstressed expression level, which was assigned with a value of 1. Data represented the average of three independent experiments ± SD. Standard errors were shown as bars above the columns. a, b, c denotes significant difference at the level of *p* < 0.05.

**Table 1 genes-10-00680-t001:** The characteristic of VvGAoxs.

Gene Name	Gene Accession No.	Position	Location	CDS (bp)	Peptide (aa)	Mw (kD)	GRAVY	pI	II	AI
*VvGA2ox1*	GSVIVT01000687001	15,497,729–15,499,650	19	972	323	36.11644	−0.193	6.54	51.22	83.90
*VvGA2ox2*	GSVIVT01000689001	15,603,207–15,604,930	19	999	333	37.29584	−0.221	8.22	35.93	88.08
*VvGA2ox3*	GSVIVT01001966001	5,864,698–5,871,371	19	990	329	37.80599	−0.304	5.85	47.92	82.64
*VvGA2ox4*	GSVIVT01010228001	17,965,019–17,966,964	1	1173	391	44.33751	−0.389	6.11	45.12	75.29
*VvGA2ox5*	GSVIVT01012628001	190,065–195,087	10	1266	422	48.52461	−0.253	6.21	49.12	80.59
*VvGA2ox6*	GSVIVT01015671001	15,307,070–15,312,521	3	1059	352	39.12278	−0.245	6.11	40.48	89.12
*VvGA2ox7*	GSVIVT01021468001	5,846,741–5,848,388	10	1002	333	37.26860	−0.288	5.42	45.89	83.33
*VvGA2ox8*	GSVIVT01028169001	4,377,203–4,380,857	7	1005	335	37.36711	−0.140	7.00	51.07	91.61
*VvGA2ox9*	GSVIVT01031814001	4,642,801–4,644,279	3	1029	343	39.34553	−0.529	5.67	30.82	81.25
*VvGA2ox10*	GSVIVT01031826001	4,723,791–4,725,151	3	1029	343	39.31948	−0.549	5.92	33.28	80.96
*VvGA2ox11*	GSVIVT01034945001	343,990–346,779	5	1020	339	37.34141	−0.213	5.40	46.47	81.36
*VvGA3ox1*	GSVIVT01008811001	2,346,915–2,348,502	18	825	275	31.27474	−0.469	5.17	26.40	83.96
*VvGA3ox2*	GSVIVT01017173001	4,993,413–4,995,290	9	1098	366	40.28499	−0.141	6.15	41.52	90.85
*VvGA3ox3*	GSVIVT01017178001	5,037,569–5,040,878	9	1080	360	39.31305	−0.050	5.14	53.05	94.31
*VvGA3ox4*	GSVIVT01020680001	3,145,880–3,152,445	12	2037	678	75.98287	−0.246	6.17	35.26	86.95
*VvGA3ox5*	GSVIVT01026928001	19,336,280–19,337,354	15	987	328	37.04838	−0.217	5.41	50.06	80.46
*VvGA3ox6*	GSVIVT01035796001	4,431,413–4,432,859	4	1068	355	39.67545	−0.175	6.46	45.63	89.55
*VvGA20ox1*	GSVIVT01008782001	1,982,673–1,985,445	18	1233	411	46.88131	−0.368	6.65	30.55	75.89
*VvGA20ox2*	GSVIVT01018453001	14,861,395–14,863,190	16	891	297	33.30088	−0.235	5.18	40.22	75.82
*VvGA20ox3*	GSVIVT01019696001	2,527,135–2,532,638	2	966	322	36.21120	−0.287	5.74	43.11	91.27
*VvGA20ox4*	GSVIVT01026453001	23,382,368–23,384,135	4	1134	378	42.94017	−0.418	6.28	33.70	77.38
*VvGA20ox5*	GSVIVT01026466001	23,110,638–23,112,150	4	1128	376	42.91282	−0.353	5.20	32.02	76.22
*VvGA20ox6*	GSVIVT01027572001	15,453,533–15,455,130	15	897	298	33.80984	−0.268	6.45	42.87	76.54
*VvGA20ox7*	GSVIVT01031837001	4,782,181–4,783,671	3	939	313	35.72351	−0.428	5.37	35.50	82.81

Notes: isoelectric point (pI), molecular weight (Mw), instability index (II), aliphatic index (AI) and grand average of hydropathicity (GRAVY), Coding sequence (CDS).

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
