# Peer review of "Genome-Wide Identification and Expression Analysis of GA2ox, GA3ox, and GA20ox Are Related to Gibberellin Oxidase Genes in Grape (Vitis vinifera L.)"

_genes, 2019, doi:10.3390/genes10090680_

Round 1

Reviewer 1 Report

General comments

Gibberellins play very important role in plant growth controlling different developmental steps related with fertility, bunch and berry traits which are of crucial importance for grapevine cultivation. The understanding of genetic basis for gibberellins synthase in grapevine is still unclear. Only few studies have reported on genomic background in grapevine in which could be a genes involved in gibberellin phenotypic expression. I would suggest to authors to shortly report and compare their results with recent findings on grape genetic basis for bunch and berry traits which could be linked with gibberellin activity such as VvGAI1 gene (see for examples articles: Vargas et al. 2013 in Euphytica; Nicolas et al. 2016 in BMC Plant Biol.; Laucou et al. 2018 in PlosOne; Guo et al. 2019 in Hort Research).

This manuscript is respectable characterization study of three gibberellin oxidase genes (GA2ox, GA3ox and GA20ox) in grape and the expression of these genes under gibberellin treatment. The authors used bioinformatics methods to identify and characterize gibberellin oxidase genes. All genomic sequences for gibberellin oxidase genes were selected from public available genome databases for Arabidopsis, Oryza, Malus and Vitis. The phylogenetic analysis showed that distribution of GA2ox, GA3ox, and GA20ox in each species was similar.

The manuscript is clearly written and the methods are accurately described. Bioinformatic methods and corresponding software’s used properly in this manuscript allowing identification of twenty-four gibberellin oxidase genes, including GA2ox, GA3ox, and GA20ox from the grape genome. In addition, qRT-PCR was used to confirm expression of GAoxs gene family under the treatment with GA3 and GA3 inhibitor (Uniconazole). The manuscript gives new insights into function and expression of gibberellin genes in grapes although study was done only on Pinot noir genotype and GA3 treatment evaluated by qRT-PCR. Further genome wide association studies involving more phenotyping data from different genotypes would needed to confirm these findings. I recommend publishing this manuscript.     

Minor comments:

In Table S8, I would suggest to replace Chinese letters into English   

Author Response

Response to Reviewers point-by-point

Dear reviewer,

Thank you for your letter and for the comments concerning our manuscript entitled “Genome-wide identification and expression analysis of GA2ox, GA3ox, and GA20ox are related to gibberellin oxidase genes in grape (Vitis vinifera L.)” (Manuscript ID: genes-559632). The comments have been valuable and have helped us revise and improve our paper. Your feedback also served as an important guide to our investigation. We have studied the comments carefully and have applied corrections, which we hope would meet your approval. The main corrections in the paper and the response to the comments are as follows:

Question1. General comments: Gibberellins play very important role in plant growth controlling different developmental steps related with fertility, bunch and berry traits which are of crucial importance for grapevine cultivation. The understanding of genetic basis for gibberellins synthase in grapevine is still unclear. Only few studies have reported on genomic background in grapevine in which could be a genes involved in gibberellin phenotypic expression. I would suggest to authors to shortly report and compare their results with recent findings on grape genetic basis for bunch and berry traits which could be linked with gibberellin activity such as VvGAI1 gene (see for examples articles: Vargas et al. 2013 in Euphytica; Nicolas et al. 2016 in BMC Plant Biol.; Laucou et al. 2018 in PlosOne; Guo et al. 2019 in Hort Research).

Answer1. We have downloaded the paper, such as Vargas et al. 2013 in Euphytica, Nicolas et al. 2016 in BMC Plant Biol., Laucou et al. 2018 in PlosOne, and Guo et al. 2019 in Hort Research. According your valuable suggestion, we have cited these papers and given a brief statement related to paper.

Question2. In Table S8, I would suggest to replace Chinese letters into English.

Answer2. We have modified all the Chinese letters into English in Table S11 (According the sequence of after modified.).

We sincerely appreciate your thoughtful comments; we believe our manuscript has been dramatically improved because of that. We hope that the changes to the manuscript will meet with approval.

Thank you and best regards.

Sincerely yours,

Honghong He [email protected]

Juan Mao    [email protected]

24 August 2019

Reviewer 2 Report

This manuscript provides an excessive amount of information about GA2ox, GA3ox and GA20ox based sequences, alignments, etc. This information is disconnected and leaves the reader wondering what is the message of manuscript. I encourage the authors to examine the Huange et al., 2015 Divergence and adaptive evolution of gibberellin oxidase gene sin plants. BMC Evolutionary Biology 207. Also, the authors present data on cis-elments in the promoter with incomplete expression data to speculate whether the cis-elements are functional. For example, the authors use previous published expression data from Grimplet et al., 2017. Where is the data showing the developmental expression profiles for the GA2, 3 and 20 oxidases?

Abstract: The abstract needs to clear and concise. There is no justification for examining the GA ox in grape. Why is this important? Also, the authors mention that GAoxs “regulate bioactive GA levels by catalyzing the later steps in the biosynthetic pathway” (page 1, line 14).. However, GA2ox GA3ox function to catabolize active GAs. Therefore, the authors need to clarify this as readers may be miss informed about the function of the GAoxs. The information from line 20 to 29, doesn’t flow and is confusing as to what the authors are addressing in the manuscript. Lastly, the authors need to provide the reader with a reasoning for performing a experiment(s) in each section, so that all the sections are linked.

Introduction:

There needs to be a better flow of information. In many cases it appears as the authors are randomly listing information regarding GA function and GAox role in plants. For example, what is the significance of listing the information Page 1, lines 38-41 in this introductory paragraph. Please modify the introduction so the readers can better understand the general role of GAs in development and stress and how the GAoxs contribute to GA function during development and stress.

Materials and methods

Page 3, line 96: What is a “tube seedling” and what is fluorescence quantification? The sentence on page 3, lines 99-102 doesn’t make sense. What material was collected for RNA extraction?

2.2-2.4. should be combined into the same section “Identification of grape gibberellin oxidase genes GA2ox, GA3ox, and GA20ox”.

2.7. cis elements are found in introns and downstream of a gene. Why did the authors only focus on cis-elements upstream of the GAoxs.

2.8. Please rewrite this section to make it clear about the gene expression data analyzed from previously published datasets. It is very confusing to read.

2.9. Where is the material methods for RNA extraction, cDNA synthesis and qRT-PCR?

Results:

3.1. Please rewrite section 3.1, as many parts are confusing such as the distribution of GAoxs on the chromosomes, size of the genes and proteins, and the properties of the GAoxs. It would help the reader if the authors provided subdivided the information and described the above for each of the GAoxs, GA20ox, GA2ox and GA3ox.

3.2. to 3.4 and discussion sections. A previous published study titled “Divergence and adaptive evolution of the gibberellin oxidase genes in plants” performed a phylogenetic and structural analyses of GA oxidases in a large set of plants, which included sequences from the alga clade and Embryophyta clade, which includes eudicots and monocots. It is really surprising that this article is not cited. I would be interested to know how the authors results compare with this 2015 BMC publication.

3.5 Codon usage…. If codon usage analysis was performed to predict whether a gene is expressed, then it would make sense to combine this data with expression data in 3.8 -3.9.

The gene duplication, triplication could be put into the structural section 3.2/3.4.

As most of the figures for this section is in the supplemental data, the detailed information in the remaining part of this section should also be put in the supplemental section.

3.6. This section should come after 3.8. This section needs to clarified as it is difficult to read. What cis-element listed indicates that a gene is expressed in a meristem? How does this cis elements listed compare the developmental profiles of these genes. The authors identify O2 sites for zein metabolism. Does grapevine produce zein?

3.7. Move to supplemental unless the authors want to demonstrate subcellular localization.

3.8 Authors need to cite the references for the gene expression data in this section. Where is the data showing the developmental expression profiles for the GA2, 3 and 20 oxidases? Why are the authors just focused on stress related expression. ?

3.9. The authors state that the expression levels of the GAox 2, 3 and 20 were examined in seedlings, but the figure indicates it was in leaves. Why are the authors only using one internal reference gene? Lastly, what is the point of this experiment?

Figure 7, I am unable to read this figure, the text is too small.

Discussion: This section needs to reworked to address questions about the possible roles of GA2, 3 and 20 oxidases in grapevine development and under stress conditions. In addition, results from grape need to be compared with structural and phylogenetic studies published by Huang et al., 2015 (BMC paper).

Author Response

Response to Reviewers point-by-point

Dear reviewer,

Thank you for your letter and for the comments concerning our manuscript entitled “Genome-wide identification and expression analysis of GA2ox, GA3ox, and GA20ox are related to gibberellin oxidase genes in grape (Vitis vinifera L.)” (Manuscript ID: genes-559632). The comments have been valuable and have helped us revise and improve our paper. Your feedback also served as an important guide to our investigation. We have studied the comments carefully and have applied corrections, which we hope would meet your approval. The main corrections in the paper and the response to the comments are as follows:

Question1. This manuscript provides an excessive amount of information about GA2ox, GA3ox and GA20ox based sequences, alignments, etc. This information is disconnected and leaves the reader wondering what is the message of manuscript. I encourage the authors to examine the Huange et al., 2015 Divergence and adaptive evolution of gibberellin oxidase gene sin plants. BMC Evolutionary Biology 207. Also, the authors present data on cis-elments in the promoter with incomplete expression data to speculate whether the cis-elements are functional. For example, the authors use previous published expression data from Grimplet et al., 2017. Where is the data showing the developmental expression profiles for the GA2, 3 and 20 oxidases?

Answer1. Thanks to your valuable comments, we have carefully read this document and conducted in-depth discussions in conjunction with our research results. We have downloaded the article Huange et al., 2015 “Divergence and adaptive evolution of gibberellin oxidase gene sin plants.” BMC Evolutionary Biology 207 and referenced the content inside to modify our article accordingly. Developmental expression profiles of GA2, 3 and 20 oxidases. After supplementation with abiotic stress expression profiles, see Figure 7 for details, the data for this section comes from the Grape eFP Browser, referenced by Fasoli, M.; Dal Santo, S.; Zenoni, S.; Tornielli, GB; Farina, L.; Zamboni, A.; Porceddu, A.; Venturini , L.; Bicego, M.; Murino, V.; Ferrarini, A.; Delledonne, M.; Pezzotti, M. The grapevine expression atlas reveals a deep transcriptome shift driving the entire plant into a maturation program. Plant Cell. 2012 , 24(9): 3489-505.

Question2. Abstract: The abstract needs to clear and concise. There is no justification for examining the GAox in grape. Why is this important? Also, the authors mention that GAoxs “regulate bioactive GA levels by catalyzing the later steps in the biosynthetic pathway” (page 1, line 14). However, GA2ox GA3ox function to catabolize active GAs. Therefore, the authors need to clarify this as readers may be miss informed about the function of the GAoxs. The information from line 20 to 29, doesn’t flow and is confusing as to what the authors are addressing in the manuscript. Lastly, the authors need to provide the reader with a reasoning for performing a experiment(s) in each section, so that all the sections are linked.

Answer2. We have rewritten the abstract. The main reason for our research is to find out how the three gibberellin oxidase genes in grapes respond to exogenous gibberellin and uniconazole, and to infer which gene can synthesize bioactive gibberellins. Which gene can degrade bioactive gibberellins? Secondly, we can predict the expression of these three gibberellin oxidase genes in various tissues of grapes and in response to abiotic stress through gene chip expression data, so as to prepare for future related research. The information from line 20 to 29 has been revised in this part. We have made detailed changes to this part.

Question3. Introduction: There needs to be a better flow of information. In many cases it appears as the authors are randomly listing information regarding GA function and GAox role in plants. For example, what is the significance of listing the information Page 1, lines 38-41 in this introductory paragraph. Please modify the introduction so the readers can better understand the general role of GAs in development and stress and how the GAoxs contribute to GA function during development and stress.

Answer3. According your suggestion, we have rewritten the introduction. We have reorganized the information regarding GA function and GAox role in plants. Also, we have modified the contents of introduction so the readers can better understand the general role of GAs in development and stress and how the GAoxs contribute to GA function during development and stress.

Question4. Materials and methods:Page 3, line 96: What is a “tube seedling” and what is fluorescence quantification? The sentence on page 3, lines 99-102 doesn’t make sense. What material was collected for RNA extraction?

Answer4. “ tube seedling” means plant tissue cultured seedlings.” fluorescence quantification” is a very important mistake, we have corrected “fluorescence quantification” into “qRT-PCR (Quantitative real-time-PCR)” and we have modified the sentence on page 3, lines 99-102. Leaf tissue of grape cv. ‘Pinot Noir’ without stem and root was used to extract RNA.

Question5. 2.2-2.4. should be combined into the same section “Identification of grape gibberellin oxidase genes GA2ox, GA3ox, and GA20ox”.

Answer5. We have combined 2.2-2.4 into the section “Identification of grape gibberellin oxidase genes GA2ox, GA3ox, and GA20ox”.

Question6. 2.7. cis elements are found in introns and downstream of a gene. Why did the authors only focus on cis-elements upstream of the GAoxs.

Answer6. Thank you very much give us a constructive suggestion. Regarding the analysis of cis-acting elements, we consulted the relevant literature according to the valuable opinions. The cis-acting elements are not only 2000bp upstream of the gene, but for most studies, many people have analyzed the upstream 2000bp, so we only made the homeopathic component of the upstream 2000bp gene, such as “Genome-wide analysis of the potato Hsp20 gene family: identification, genomic organization and expression profiles in response to heat stress” Zhao et al., 2018. BMC Genomics, 19:61. In the future study, we will modify our knowledge to explore the cis-elements.

We identify O2 sites for zein metabolism, but this is not very relevant to the grape, so we removed it in this revision, and the figure have been redone(Figure 7).

Question7. 2.8. Please rewrite this section to make it clear about the gene expression data analyzed from previously published datasets. It is very confusing to read.

Answer7. We have rewritten this section and this part mainly explains the results of our research. As for the results of the previous research and our research results, the discussion is mainly in the discussion.

Question8. Where is the material methods for RNA extraction, cDNA synthesis and qRT-PCR?

Answer8. We have added RNA extraction, cDNA synthesis and qRT-PCR methods in 2.7.

Question9. Results: 3.1. Please rewrite section 3.1, as many parts are confusing such as the distribution of GAoxs on the chromosomes, size of the genes and proteins, and the properties of the GAoxs. It would help the reader if the authors provided subdivided the information and described the above for each of the GAoxs, GA20ox, GA2ox and GA3ox.

Answer9. We have rewritten ‘the Results: 3.1’and added the subdivided information of GAoxs, GA20ox, GA2ox and GA3ox.

Question10. 3.2. to 3.4 and discussion sections. A previous published study titled “Divergence and adaptive evolution of the gibberellin oxidase genes in plants” performed a phylogenetic and structural analyses of GA oxidases in a large set of plants, which included sequences from the alga clade and Embryophyta clade, which includes eudicots and monocots. It is really surprising that this article is not cited. I would be interested to know how the authors results compare with this 2015 BMC publication.

Answer10. Thanks to the valuable comments. For the part of 3.2-3.4, we have made corresponding modifications according to the research results of Huang et al. 2015.

Question11. 3.5 Codon usage…. If codon usage analysis was performed to predict whether a gene is expressed, then it would make sense to combine this data with expression data in 3.8 -3.9. The gene duplication, triplication have been put into the structural section 3.2/3.4. As most of the figures for this section is in the supplemental data, the detailed information in the remaining part of this section should also be put in the supplemental section.

Answer11. Thanks to the valuable comments. We have combined the codon preference with the gene expression for detailed analysis based on the reviewer’s opinion. The gene duplication and triplication have been putted into the supplemental section.

Question12. 3.6. This section should come after 3.8. This section needs to clarified as it is difficult to read. What cis-element listed indicates that a gene is expressed in a meristem? How does this cis elements listed compare the developmental profiles of these genes. The authors identify O2 sites for zein metabolism. Does grapevine produce zein?

Answer12. Thank you very much for pointing out our mistakes, we have removed 3.6 after 3.8. Regarding the analysis of cis-acting elements, we consulted the relevant literature according to the valuable opinions. The cis-acting elements are not only 2000bp upstream of the gene, but for most studies, many people have analyzed the upstream 2000bp, so we only made the homeopathic component of the upstream 2000bp gene, such as “Genome-wide analysis of the potato Hsp20 gene family: identification, genomic organization and expression profiles in response to heat stress” Zhao et al., 2018. BMC Genomics, 19:61. We identify O2 sites for zein metabolism, but this is not very relevant to the grape, so we removed it in this revision, and the figure have been redone(Figure 7).

Question13. 3.7. Move to supplemental unless the authors want to demonstrate subcellular localization.

Answer13. We have moved subcellular localization to the supplemental.

Question14. 3.8. Authors need to cite the references for the gene expression data in this section. Where is the data showing the developmental expression profiles for the GA2, 3 and 20 oxidases? Why are the authors just focused on stress related expression. ?

Answer14. The chip data on the expression of grape tissue has been explored before, and the content of the organization expression database has only recently been studied. We have added it in this revision, see Fig. 6.

Question15. 3.9. The authors state that the expression levels of the GAox 2, 3 and 20 were examined in seedlings, but the figure indicates it was in leaves. Why are the authors only using one internal reference gene? Lastly, what is the point of this experiment?

Answer15. When the grapes are subjected to abiotic stress, the most sensitive part is the leaf tissue, so the main material we choose for qRT-PCR analysis is the grape leaf tissue. GAPDH gene (GenBank accession no. CB973647) was used as the internal reference gene, GAPDH or G3PDH is the abbreviation of glyceraldehyde-3-phosphate dehydrogenase. GAPDH is an enzyme in the glycolysis reaction that is widely distributed in cells in various tissues.

Question16. Figure 7, I am unable to read this figure, the text is too small.

Answer16. Figure 7 have modified to Figure 8, and the font was modified in aim to better readability.

Question17. Discussion: This section needs to reworked to address questions about the possible roles of GA2, 3 and 20 oxidases in grapevine development and under stress conditions. In addition, results from grape need to be compared with structural and phylogenetic studies published by Huang et al., 2015 (BMC paper).

Answer17. We have rewritten the discussion section. In combination with the comments from the reviewer, we have condensed the entire experimental results and discussed the results in depth with the relevant literature.

We sincerely appreciate your thoughtful comments; we believe our manuscript has been dramatically improved because of that. We hope that the changes to the manuscript will meet with approval.

Thank you and best regards.

Sincerely yours,

Honghong He [email protected]

Juan Mao    [email protected]

24 August 2019
